# Experimental Basis Sets of Quantification of Brain ^1^H-Magnetic Resonance Spectroscopy at 3.0 T

**DOI:** 10.3390/metabo13030368

**Published:** 2023-03-01

**Authors:** Hyeon-Man Baek

**Affiliations:** 1Department of Health Sciences and Technology, GAIHST, Gachon University, Incheon 21999, Republic of Korea; hmbaek98@gachon.ac.kr; Tel.: +82-32-899-6678; 2Department of Molecular Medicine, Lee Gil Ya Cancer and Diabetes Institute, Gachon University, Incheon 21999, Republic of Korea

**Keywords:** MRS, metabolite, quantification, LCModel, basis set

## Abstract

In vivo short echo time (TE) proton magnetic resonance spectroscopy (^1^H-MRS) is a useful method for the quantification of human brain metabolites. The purpose of this study was to evaluate the performance of an in-house, experimentally measured basis set and compare it with the performance of a vendor-provided basis set. A 3T clinical scanner with 32-channel receive-only phased array head coil was used to generate 16 brain metabolites for the metabolite basis set. For voxel localization, point-resolved spin-echo sequence (PRESS) was used with volume of interest (VOI) positioned at the center of the phantoms. Two different basis sets were subjected to linear combination of model spectra of metabolite solutions in vitro (LCModel) analysis to evaluate the in-house acquired in vivo ^1^H-MR spectra from the left prefrontal cortex of 22 healthy subjects. To evaluate the performance of the two basis sets, the Cramer-Rao lower bounds (CRLBs) of each basis set were compared. The LCModel quantified the following metabolites and macromolecules: alanine (Ala), aspartate (Asp), γ-amino butyric acid (GABA), glucose (Glc), glutamine (Gln), glutamate (Glu), glutathione (GHS), Ins (myo-Inositol), lactate (Lac), N-acetylaspartate (NAA), N-acetylaspartylglutamate (NAAG), taurine (Tau), phosphoryl-choline + glycerol-phosphoryl-choline (tCho), N-acetylaspartate + N-acetylaspartylglutamate (tNA), creatine + phosphocreatine (tCr), Glu + Gln (Glx) and Lip13a, Lip13b, Lip09, MM09, Lip20, MM20, MM12, MM14, MM17, Lip13a + Lip13b, MM14 + Lip13a + Lip13b + MM12, MM09 + Lip09, MM20 + Lip20. Statistical analysis showed significantly different CRLBs: Asp, GABA, Gln, GSH, Ins, Lac, NAA, NAAG, Tau, tCho, tNA, Glx, MM20, MM20 + Lip20 (*p* < 0.001), tCr, MM12, MM17 (*p* < 0.01), and Lip20 (*p* < 0.05). The estimated ratio of cerebrospinal fluid (CSF) in the region of interest was calculated to be about 5%. Fitting performances are better, for the most part, with the in-house basis set, which is more precise than the vendor-provided basis set. In particular, Asp is expected to have reliable CRLB (<30%) at high field (e.g., 3T) in the left prefrontal cortex of human brain. The quantification of Asp was difficult, due to the inaccuracy of Asp fitting with the vendor-provided basis set.

## 1. Introduction

In vivo proton magnetic resonance spectroscopy (^1^H-MRS) is a non-invasive technique that can provide neurochemical information through the quantification of metabolite concentrations. This technique has been used as a tool to study not only metabolic disorders in patients, but also metabolic changes in healthy people [1,2,3,4,5]. An in-depth understanding of metabolic processes is crucial in studying diseases or monitoring treatment effects. Accurate detection and estimation of changes in metabolite concentrations are necessary to understand metabolic processes, especially in the brain [6]. However, accurate and precise metabolite quantification, especially at short echo times (TE), is difficult because it is not easy to separate overlaid complex metabolite signals, such as those from γ-amino butyric acid (GABA), glutamate (Glu), glutamine (Gln), aspartate (Asp), glutathione (GSH), and myo-inositol (Ins), due to their inherent low concentrations in the brain [7].

Quantification involves the determination of metabolite concentrations from non-processed magnetic resonance spectra. For objective, accurate, and reliable quantification analysis, quantification procedures with automatic algorithms have been developed to minimize user interactions [8,9,10]. The automatic analysis method has the advantage of being more precise and objective because it does not depend on user abilities. LCModel (linear combination of model spectra of metabolite solutions in vitro) is a widely used algorithm that uses each metabolite spectrum pattern to estimate metabolite concentrations [8]. This algorithm needs a basis set that is composed of individual metabolite spectra for the quantification of acquired signals. The basis set is fitted to the acquired spectrum. LCModel uses a spline function to model a baseline composed of macromolecules and lipid signals. It has been reported that this method can provide improved accuracy and precision for signals such as Glu, Gln, and lipids over methods such as the advanced method for accurate, robust, and efficient spectral fitting (AMARES) [11,12]. Recently, other methods, such as QUantum ESTimation (QUEST), accurate quantitation of short echo time domain signals (AQSES), and totally automated robust quantitation in NMR (TARQUIN), were developed, which also use a metabolite basis set [9,10,13].

Basis sets can be generated through experimental measurement or theoretical simulation. To generate an experimental basis set, phantoms need to be developed for each metabolite with reference chemicals, while temperature and pH need to be adjusted to simulate physiological conditions [7,14]. After developing individual phantoms, each phantom is scanned using an appropriate echo time and pulse sequence that can subsequently be applied to in vivo human data [15,16]. A simulated basis set is generated computationally using the prior knowledge of physical and chemical characteristics of each metabolite and pulse sequence parameters. The simulation is made based on density matrix formulation [17,18,19]. The simulation method can be used to generate a basis set with greater speed and ease than the experimental method. With information on pulse sequence parameters and in vivo experimental conditions, each simulated basis set can be generated quickly and applied easily to specific experiments [15]. However, the method assumes an ideal pulse sequence and cannot describe the exact experimental conditions. Thus, basis sets developed for different conditions would yield different quantification results.

Previous studies have quantitatively compared the performances between simulated and experimentally measured basis sets [15,16]. These studies concluded that quantification differences between simulated and experimental basis sets are not significant, and they can be interchangeable. Cudalbu et al. [15] used the QUEST algorithm to analyze spectra. The QUEST algorithm also uses a metabolite basis set as prior knowledge. However, unlike the LCModel basis set, which is known to permit more flexible line shapes, the basis set in QUEST is Lorentzian line broadening. The use of a measured basis set might result in the loss of information because they have more similar line shapes to the in vivo spectra before line broadening. Therefore, the usage of the measured basis set might not fully reflect the experimental conditions. Wilson et al. [16] compared the difference between the experimentally and theoretically acquired basis sets. In the study, they used the LCModel for spectral analysis. However, they used an experimental basis set provided by a vendor. They applied the basis set that was not acquired in the same environment to the in vivo data. Therefore, the advantages of experimentally generated basis sets for quantification observed in the previous studies might not have been reflected fully.

In this study, we acquired an experimental basis set and performed in vivo ^1^H-MRS using an in-house magnetic resonance imaging (MRI) system. In addition, we fit the in vivo data with the experimental and vendor-provided basis sets to test performance of the basis sets. The spectrum quality was investigated to minimize fitting instabilities caused by poor quality data. After LCModel spectral fitting analysis, we investigated the differences of corrected metabolite concentrations and Cramer-Rao lower bounds (CRLBs) between the two approaches. The correlation and coefficient of variations were also investigated to test their consistency between the two approaches.

## 2. Materials and Methods

### 2.1. Generating Experimental Basis Set

A total of 16 metabolite samples (e.g., alanine (Ala), N-acetylaspartate (NAA), N-acetylaspartylglutamate (NAAG), creatine(Cr), phosphocreatine (PCr), phosphoryl-choline (PCh), glycerol-phosphoryl-choline (GPC), taurine (Tau), Glu, Gln, GSH, Ins, GABA, Asp, lactate (Lac), and glucose (Glc)) were purchased from Sigma-Aldrich (Sigma-Aldrich Korea Ltd. (Seoul, Republic of Korea), SIAL) and used to generate the experimental basis set. Types and quantities of metabolites used in the study were determined according to the LCModel manual (http://s-provencher.com/lcm-manual, accessed on 1 November 2013). Each metabolite was dissolved in a standard chemical solvent. When using NaOH and HCl, solvent pH was adjusted to 7.2 and experiments were performed at room temperature.

A 32-channel, receive-only phased array head coil was used to perform ^1^H-MRS analysis of the metabolite phantoms. Experiments were conducted using a 3T (Philips Achieva System 3.0 TX, Best, The Netherlands, Release 3.2.3). For voxel localization, point-resolved spin-echo sequence (PRESS) [18] was used, and volume of interest (VOI) was positioned at the center of phantoms. VOI size was 20 × 20 × 20 mm^3^. For water suppression, variable pulse power and optimized relaxation delays (VAPOR) pulse [19] were used. Repetition time (TR)/echo time (TE) were 10,000/35 ms, and number of excitations (NEX) was 128. A total of 16 metabolite spectra were imported to the LCModel and used to generate the experimental basis set.

### 2.2. In Vivo MRI and ^1^H-MRS Study

In vivo ^1^H-MRS experiments were performed using the same MRI system used in the phantom experiments to acquire measured basis sets. After the localizer scan, three orthogonal images using T_2_-weighted spin echo sequence (TR/TE = 2500/80 ms) were used, placing the volume of interest in the brain before ^1^H-MRS. Localized single-voxel ^1^H MR spectra were acquired from the left prefrontal cortex of 22 healthy subjects in this study. Acquisition parameters were as follows: PRESS pulse sequence for voxel localization, TR/TE = 3000/35 ms, voxel size = 20 × 20 × 20 mm^3^, 128 repetitions for averaging, 2048 data points, bandwidth = 2000 Hz. RF pulse shapes used in the PRESS were “spredrex” (bandwidth/duration, 3877 Hz/4.20 ms) and “gts1203” (bandwidth/duration, 2289 Hz/4.07 ms) for excitation and refocusing. First-order iterative VOI shimming method was performed automatically on the water resonance for optimization of the homogeneities in each VOI. After shimming procedure, VAPOR pulse [19] was applied for water suppression. We turned on the spectral correction when acquiring each spectrum. Sixteen fully relaxed water unsuppressed signals were also acquired to measure the water reference. This study protocol was approved by the University of Chungbuk Institutional Review Board (#CBNU-201506-BMSBBR-059-01), and informed consent was obtained from each patient.

### 2.3. In Vitro ^1^H-MRS Phantom Study

A standard GE MRS phantom (12.5 mM NAA, 10 mM Cr, 3 mM Cho, 12.5 mM Glu, 7.5 mM mI, and 5 mM Lac) was included in the study. Then, both sets were applied to identify whether either approach is more precise or accurate. We used same PRESS pulse sequence for voxel localization, and acquisition parameters were TR/TE = 10,000/35 ms, voxel size = 25 × 25 × 25 mm^3^, 128 repetitions for averaging, 2048 data points, and bandwidth = 2000 Hz.

### 2.4. Quality Control of ^1^H-MRS Spectra

Spectral quality analysis was performed using our own analysis program written in Matlab (Version 6.5 for Windows, The MathWorks, Inc., Natick, MA, USA) on a PC, as a standard of judgment for further quantification. Before spectral quality estimation, zero-order phase correction was performed to correct overall distortion. Spectral quality was estimated using the full width at half-maximum (FWHM) of unsuppressed water signal and the signal-to-noise ratio (SNR) of the suppressed signal. After Lorentzian line fitting with the water signal, the FWHM was measured. Peak height of the highest signal between 0 to 3.4 ppm and standard deviation of noise between 9 to 11 ppm were measured, and SNR was estimated. In addition to FWHM and SNR, extent of water suppression was estimated by measuring before and after heights of the water peak. Data would be excluded if the FWHM was higher than 8 Hz or SNR was lower than 10. These criteria were adopted from [20].

### 2.5. LCModel Spectral Analysis

The LCModel algorithm was used for quantitative analysis of spectra from 22 healthy subjects. The LCModel employs semi-parametric non-linear least square (NLLS) approaches for spectral analysis [8,21]. It uses neurochemical information of metabolites as prior knowledge and employs a purely mathematical method. The signal model used in the LCModel is based on Fourier transformed data in the frequency domain and composed of mainly two parts. One part describes the background signal of the spectrum, while the other part represents the line shapes of various metabolite signals. The objective of the algorithm is to identify parameters to minimize cost function defined in the model. This minimization process is known to be performed by a type of regularized constrained nonlinear least square (CONTIN) method that was developed by [22]. Background in the spectrum is modeled using a cubic B-spline function, and metabolites signals are modeled by acquiring in vitro measured or simulated metabolite basis sets [21]. Spectral fitting is performed automatically with minimum user interaction. After analysis, estimation of fitting error is represented by CRLB, and quantification result of metabolite concentrations is also given.

### 2.6. Quantification of Metabolite Concentrations and CSF Correction

For the water scaling method, metabolite concentration in the LCModel was calculated using the following equation:(1)$${\mathrm{Conc}}_{\mathrm{met}}=\left({\mathrm{Ratio}}_{\mathrm{area}}\right)\;\times \;\frac{2}{N1H_{\mathrm{met}}}\;\times \;\frac{{\mathrm{ATT}}_{H2O}}{{\mathrm{ATT}}_{\mathrm{met}}}\;\times \;W_{\mathrm{conc}}$$
where Ratio_area_ indicates the ratio of water peak area in the unsuppressed signal to each metabolite peak area in the suppressed spectrum. $N1H_{\mathrm{met}}$ indicates the number of protons that contribute to resonance. For example, for a CH_2_ group, N1H_met_ is 2. W_conc_ is the water concentration in the VOI, the default value in the LCModel is 35,880 mmol/L. ATT_H2O_ and ATT_met_ are parameters that represent attenuation of signals. ATT_H2O_ is set as 0.7 as default [14,23].

On the other hand, VOI for in vivo ^1^H-MRS includes not only gray matter and white matter tissues, but also contains cerebrospinal fluid (CSF), in general. Metabolites that we aimed to measure in the study are contained in the brain tissue and not in the CSF. Therefore, the estimated metabolite concentrations are much lower than the true values if the CSF is not considered in the VOI. For more accurate quantification, we performed CSF correction. For estimation of CSF percentage in the VOI for each subject, statistical parametric mapping (SPM) was used for segmentation [24]. T2-weighted images were segmented into gray matter, white matter, and CSF. After brain segmentation, signals exclusively in the VOI of each segmented image were extracted using mask [25]. In addition, using Otsu’s method, the threshold value was determined to classify the intensity distribution [26]. Intensities higher than the threshold were considered tissue, and intensities lower than threshold value were considered background. Percentage of CSF could be estimated as below.
(2)$${\mathrm{CSF}}_{\mathrm{VOI}}=\frac{N_{\mathrm{CSF}}}{N_{\mathrm{VOI}}}$$
where ${\mathrm{CSF}}_{\mathrm{VOI}}$ is CSF ratio in the VOI, $N_{\mathrm{CSF}}$ is number of voxels determined as CSF, and $N_{\mathrm{VOI}}$ is total number of voxels in the VOI. Thus, metabolite concentrations estimated by the LCModel were corrected [27].
(3)$${\mathrm{Conc}}_{\mathrm{corrected}}={\mathrm{Conc}}_{\mathrm{measured}}\;\times \;\frac{1}{1-{\mathrm{CSF}}_{\mathrm{VOI}}}$$
where ${\mathrm{Conc}}_{\mathrm{corrected}}$ is the corrected metabolite concentration, ${\mathrm{Conc}}_{\mathrm{measured}}$ is the metabolite concentration given by LCModel, and ${\mathrm{CSF}}_{\mathrm{VOI}}$ is estimated CSF ratio in the VOI.

### 2.7. Statistical Analysis

To compare the spectral fitting performance of two different basis sets, the Shapiro–Wilk test for normality, paired *t*-test, and paired Wilcoxon Rank Sum test (or Mann–Whitney U test) were performed using estimated concentrations and CRLBs of two approaches. To investigate which metabolites or macromolecules can be separated or not, we calculated cross-correlations for metabolites and macromolecules. The coefficient of variation was also investigated/ and its relation to the mean CRLB (relative units) was examined [28]. All statistical analysis was performed by R software (https://www.r-project.org/, accessed on 1 October 2022).

## 3. Results

### 3.1. Generating Experimental Basis Set

Figure 1 shows a total of 16 types of spectra of metabolites. From bottom to top, each spectrum indicates Ala, Asp, Cr, PCr, GABA, Glc, Gln, Glu, GSH, GPC, PCh, Ins, Lac, NAA, NAAG, and Tau. We can also identify reference peaks, such as (3-trimethylsilyl)-1-propane-sulfonic acid (DSS) and formate at 0 and 8.4 ppm, respectively. Formate is necessary to scale for each metabolite spectrum, and DSS is needed to identify the peak position of other metabolites from reference. The residual water signal after water suppression is shown at 4.8 ppm.

### 3.2. Quality Control of ^1^H-MRS Spectra

Figure 2 shows that the spectral quality of the 22 subjects satisfies the criteria (FWHM < 8 Hz, SNR > 10). The horizontal axis represents the FWHM of the water peak before water suppression, and the vertical axis represents the SNR after water suppression. SNR is the maximum signal intensity in the 0–3.4 ppm region, with respect to the standard deviation of the noise in the 9–11 ppm range. The FWHMs of 22 spectra were in the 8-Hz range. Therefore, shimming in the VOI was considered successful. If the shimming procedure is not well-performed, the width of the peaks in the spectra is widened, resulting in a greater overlap of the metabolite peaks, which makes distinguishing metabolite peaks more difficult. It can also result in poor fitting accuracy and eventually make the quantification of metabolite concentrations more difficult. Results from 22 spectra revealed that all measured SNRs were greater than 10. This indicates that peaks for metabolites such as NAA, tCr, and tCho are 10 times greater in magnitude than the noise. After zero-order phase correction, water peak heights before and after water suppression were measured. Results were 0.7 ± 0.4 (mean ± SD) %, less than 1%, indicating good water suppression.

### 3.3. LCModel Spectral Fitting Analysis

Figure 3 shows the mean spectrum of 22 spectra. Figure 3a shows a spectral analysis result using the basis set obtained experimentally on the same system. Figure 3b shows a spectral analysis result using the vendor provided basis set, which was not generated from the experimental basis set and in vivo data. The black line in Figure 3 indicates phased in vivo data, and the red line indicates fitted data. The lines below the red line show the baseline and each fitted metabolite signal. The blue line at the top of the figure represents the fitting residues of the difference between in vivo and fitted data. The Shapiro–Wilk normality test of fitting residues showed that using the in-house experimental basis set (0 of 22, *p* < 0.05) was better than using the provided basis set fitting (6 of 22, *p* < 0.05).

Table 1 shows the mean CRLBs and standard errors of each of the metabolites and macromolecules. The result indicates that fitting precisions using the in-house experimental basis set were better than the provided simulated basis set for Asp, GABA, Gln, GSH, Ins, Lac, NAA, NAAG, Tau, tCho, tNA, Glx (*p* < 0.001), and MM12 (*p* < 0.01). On the other hand, there were opposite results for tCr (*p* < 0.01), Lip20 (*p* < 0.05), MM20 (*p* < 0.001), MM17 (*p* < 0.01), and MM20 + Lip20 (*p* < 0.001), and there were no precision differences for Ala, Glc, Glu, Lip13a, Lip13b, Lip09, MM09, MM14, Lip13a + Lip13b, MM14 + Lip13a + Lip13b + MM12, and MM09 + Lip09.

Table 2 shows the mean concentrations and standard errors of each of the metabolites and macromolecules. The paired *t*-test indicates that quantified concentrations using the in-house experimental basis set are significantly lower than the provided simulated basis set for GABA, Gln, GSH, Ins, Lac, NAA, NAAG, Tau, tCho, tNA, Lip09, and MM17 (*p* < 0.001). The opposite results were found for Asp, Glu, Glx, tCr, Lip20, MM20, MM12, MM20 + lip20 (*p* < 0.001), MM09, and MM14 + Lip13a + Lip13b + MM12 (*p* < 0.05). Additionally, there were no significant differences for Ala, Glc, Lip13a, Lip13b, MM14, Lip13a + Lip13b, and MM09 + Lip09.

### 3.4. Correlation Matrices

Figure 4 shows the correlation matrices of metabolite and macromolecules concentrations, as quantified using both basis sets basis. The blue colors indicate positive correlations, red colors indicate negative correlations, and blanks indicate no significant correlations. Before thresholding, Figure 4a (left, the in-house experimental basis set) and Figure 4b (right, the provided simulated basis set) showed very similar patterns, and we could not find any noticeable differences between the two matrices; thus, we adjusted significant level of the P value to 0.1. There were negative correlations in the left matrix between GABA and MM09, GABA and MM12, Glc and NAA, Glc and tNA, Glc and Lip09, Gln and MM17, Lac and Lip13b, and Lac and Lip13a + Lip13b. On the other hand, there were inverse correlations in the right matrix between Asp and NAAG (*p* = 0.051), Gln and MM14, Gln and MM17, Lac and Lip13a, Lac and Lip13b, Lac and Lip13a + Lip13b, Lac and MM14 + Lip13a + Lip13b + MM12, tCr and Lip13b.

### 3.5. In Vitro ^1^H-MRS Phantom Study

Table 3 shows a quantification result of standard MRS phantom. Metabolite concentrations were underestimated from true concentrations (Ins; -1.4 mM, Lac; -1.63 mM, Cho; −0.55 mM, NAA; −0.92 mM, Cr; −0.15 mM, and Glu; −0.15 mM) using the in-house experimental basis set. On the other hand, concentrations were overestimated (Ins; +0.39 mM, Cho; +0.54 mM, NAA; +0.83 mM, and Cr; +0.37 mM) and underestimated (Lac; −0.67 mM, and Glu; −0.74 mM). We could not find any notable differences in relative units. However, in an absolute unit, we could find that the experimental basis set was more precise for Ins, Lac, Cho, NAA, and Glu, but not for Cr, than the provided simulated basis set.

### 3.6. Coefficient of Variation

To investigate the different variations of the results using the two different approaches, coefficients of variation (CV) were compared. CV is defined as metabolite concentration divided by its standard deviation. Figure 5 shows CV versus mean CRLBs for the two approaches. We use CRLBs in relative unit (i.e., %SD) to compare with the study and to interpret the results. We included all subjects’ data. CVs of metabolites for which CRLBs are less than 15%: Glu (10.1 vs. 7.6%, the in-house experimental basis set vs. the provided simulated basis set), Ins (12.4 vs. 12.3%), NAA (18.7 vs. 7.2%), tCho (20.5 vs. 15.5%), tNA (14.1 vs. 20.1%), tCr (18.1 vs. 16.4%), and Glx (7.4 vs. 9.0%). The metabolites appeared above the identity line in Figure 5b. The CVs for which CRLBs are less than 35% include: Gln (26.2 vs. 25.0%), GSH (12.3 vs. 12.7%), MM09 (10.7 vs. 9.9%), MM20 (9.4 vs. 9.6%), MM12 (11.8 vs. 11.3%), MM14 (15.5 vs. 11.9%), MM17 (12.6 vs. 19.4%), MM14 + Lip13a + Lip13b + MM12 (10.7 vs. 9.3%), MM09 + Lip09 (6.8 vs. 7.2%), and MM20 + Lip20 (10.0 vs. 10.5%). The metabolites and macromolecules appear below the identity line in Figure 5b. The CVs for which CRLBs are greater than 35% include: GABA (125.5 vs. 47%), NAAG (192.6 vs. 146.2%), and Lac (141.6 vs. 134.2%). The metabolites appear above the identity line in Figure 5a. However, Asp (25.0 vs. 155.2%) and Tau (49.5 vs. 28.6%) appeared in different regions from each other.

## 4. Discussion

In this study, we aimed to evaluate the performance of the in-house experimental basis set, compared to the vendor, provided simulated basis set by applying in-house acquired in vivo ^1^H-MRS data. Data were acquired from the left prefrontal cortex of 22 normal subjects. Fitting errors from two approaches using CRLBs were compared. We found that the LCModel quantification with the in-house experimental basis set had significantly lower CRLBs than the provided simulated basis set for most metabolites and macromolecules at 3T.

According to Wilson et al. [16], there are two main reasons for using different basis sets. First, metabolite signals can be different for each basis set for several reasons, including signal amplitude, SNR, or baseline. This type of difference is referred to as differences in basis sets. Second, the fitting algorithms may cover different solutions. In this case, the fitting residue can be similar, but with different quantification results. For example, high PC and low GPC concentrations may yield similar results as low PC and high GPC. These differences are referred to as fitting instability. When using the two different basis sets, fitting instabilities are readily apparent. Therefore, to measure the difference between basis sets accurately, it is important to reduce the fitting algorithm-dependent fitting instabilities as much as possible. Discrepancy due to differences in basis sets should be independent of the quality of data. Wilson et al. [16] suggested that a level of agreement of ±0.99 mM between simulated and experimental basis sets is sufficiently small for them to be used interchangeably. While small fitting instabilities are inevitable, it should not be noticeable for high quality data. They reported that fitting instabilities are dominant in poor quality data because the poor quality results in a loss of intrinsic information.

Graaf et al. [20] made protocols for spectrum quality assessment and applied it to their study. The protocol contains investigations of FWHM (<8 Hz), SNR (>10), and artifacts. The thresholds of FWHM and SNR for quality assessment were determined by experts. This protocol was more conservative than the criteria determined by a supervised pattern recognition method. In our study, the results satisfied the above-mentioned protocol (e.g., FWHM < 8 Hz, SNR > 10, no artifact). The difference is B_0_ field strength. The Graaf et al. [20] study was performed at 1.5T, while we experimented at 3T. The increased strength of the magnetic field linearly correlates with SNR; our results show a larger SNR value using a 3T magnetic field than the SNR value of 20 from the experiment that used a 1.5T magnetic field. On the other hand, if the field strength is increased, the local field inhomogeneity is increased, and it makes T2 relaxation time shorter. Therefore, the line width of spectral peaks is broadened, and FWHM is increased. If SNR is low, it is difficult to distinguish between metabolite peaks and noise, and fitting instabilities will increase, making the quantification of metabolite concentrations difficult. If water suppression is not adequate, water peaks hinder the fitting of other metabolites, making quantification difficult. In our study, FWHM, SNR, and the extent of water suppression show that these parameters do not have a significant impact on quantitative analysis (Figure 2).

Differences between the fitting errors of the two approaches can be due to the following reasons. The RF pulse shapes used to acquire the two basis sets are different for the two approaches. The basis sets used in this study were in-house experimentally generated and received from a vendor. Both basis sets were acquired using the PRESS sequence, and TE was 35 ms. The in-house experimental basis set was generated using the same parameters used in in vivo and standard MRS phantom experiments. A pulse diagram contains RF pulses (excitation and refocusing) that were taken in real time. The pulse diagram also contains various gradients (slice selection and spoiler) that have finite ramp times. MRI machine might have had delayed times, turning on the pulses resulting in an imperfect shimming. Therefore, magnetic field inhomogeneity during real experiments is inherent. On the other hand, the provided simulated basis set assumes ideal hard pulses and does not consider localization, pulse offsets, pulse power, and pulse lengths, which results in two different basis sets.

Kaiser et al. [29] showed a comparison of ideal and 3D-localised PRESS simulations with experimental 3D PRESS-localized phantom spectra of Lac, NAA, Glu, and Ins at different field strengths. They reported that PRESS-localized spectra contained discrepancies between ideal and localized conditions. They also demonstrated that those discrepancies are more severe in higher fields. As such, the difference in the performance of two basis sets and the discrepancies in spectral line shape may be due to the inconsistency of localization conditions. This difference is expected to be more apparent for the strongly coupled metabolites because of their complex appearances. Compared to weakly coupled or non-coupled metabolites, strongly coupled metabolites can result in a more noticeable difference [30]. We investigated regularization parameters (e.g., $\alpha_{S}$ and $\alpha_{B}$ in the LCModel method (Provencher [21])) to identify the localization effects from our data. The output parameters $\alpha_{S}$ (8.4 ± 19.08 vs. 62.5 ± 77.0, mean ± SD), $\alpha_{B}$ (0.065 ± 0.02 vs. 0.073 ± 0.03), and $\alpha_{S}/\alpha_{B}$ (129 ± 283.15 vs. 856 ± 1572.06) were significantly lower with in-house experimental basis set (*p* < 0.05). Therefore, we believe that fitting with the provided simulated basis set needed more regularization terms to account for line shape discrepancies than with in-house experimental basis set fitting, in order to minimize the objective function.

Recently, Deelchand et al. [30] performed a simulation study to investigate how the quantitative accuracy of the metabolite can be increased by using a short echo time when the B_0_ field increases in the human brain. They reported that greater improvement in quantification precision is obtained for J-coupled metabolites than for singlets as the B_0_ increases. They also reported that additional improvement of quantification precision is expected for J-coupled metabolites because of relatively simple spectral patterns, which can be easily distinguished by reducing the peak overlap. They suggested the possibility of quantifying metabolites, such as NAA, tCho, Ins, Glx, tCr, and Cr, reliably (CRLB < 25%) at 3T. In our study, we could also quantify GSH, Glu (<25%), Gln, and Asp (<30%). According to Deelchand et al. [30], CRLBs at 7T are decreased inverse proportionally to square root of B_0_, we can expect improved precision for GSH, Glu, Gln, and Asp in the VOI due to well-separated peaks.

In this study, the estimated ratio of CSF in the region of interest was calculated to be about 5%. The concentration of each metabolite slightly increased after correction. We compared our results with those of other studies, such as Horder et al. [5], which had results that were very similar to our study’s. They performed their experiment in the prefrontal cortex area. They estimated the CSF in the area using SPM segmentation and utilized LCModel analysis, but their experiments were carried out using 1.5T. They reported the metabolite concentrations (mean ± SD, mM) for Glx (8.0 ± 1.4), tCho (1.3 ± 0.3), tCr (4.6 ± 0.6), tNA (6.7 ± 1.0), and CSF percentage were 4.03 ± 3.58%. In our study, the CSF corrected metabolite concentrations were for Glx (the in-house experimental basis set; 8.2 ± 0.7, the provided simulated basis set; 8.0 ± 0.6), tCho (1.6 ± 0.1 vs. 1.6 ± 0.1), tCr (5.3 ± 0.4 vs. 5.0 ± 0.4), and tNA (6.8 ± 0.5 vs. 7.6 ± 0.6). CSF (5 ± 2%) was a little higher in our results (Table 2). These differences (mean ± SD, years) might have been due to the control group age difference between the two studies (34 ± 8.8 vs. 52 ± 8.4 years). The tNA concentration with the in-house experimental basis set was more similar. Therefore, the result indicates that the provided simulated basis set overestimates a tNA concentration at 3T.

We could not find any noticeable differences between the two matrices of metabolite and macromolecules concentrations quantified using both basis sets (Figure 4). It is expected that Asp and NAAG can be quantified individually using the in-house experimental basis set from the correlation analysis. GABA, Gln, and Lac showed negative correlations with lipids and macromolecules. Other metabolites had no significant negative correlations for both basis sets. It can be interpreted as individual quantifications being possible. However, we note that concentrations of Ala, Glc, and Lip13b are quantified as zero in our data. On the other hand, we observed a significant inverse correlation between Asp and NAAG (e.g., Asp = 0 mM, NAAG = 0.978 mM), which indicates that the provided simulated basis set was unable to separate these two metabolites. This spectrum is well-followed by fit lines for both basis sets in the 2.5–2.9 ppm overlapped region.

Deelchand et al. [28] reported the dominant coefficient of variation (CV), when compared with the CRLB (<15%), indicating that it is feasible to compare concentrations between subjects. Conversely, if CRLB is superior to the between-subjects CV, it indicates that the measurement error is dominant over the difference between the physiological effects of subjects, which means that comparative measurement errors might be possible. The results showed that the CV of NAA, tNAA, tCr, tCho, Ins, Glu, and Glc + Tau is dominant over their CRLB. In the case of our results, the CV of NAA, tNAA, tCr, tCho, Ins, Glu, and Glx was dominant over CRLB (Figure 5).

Our limitation is that we could not explain the background (or baseline), which constitutes about 50% of the spectrum. The background consists of macromolecules, which have very short T1, T2 relaxation times, and broad FWHM. Therefore, background estimation affects metabolite quantification. In this study, we used the LCModel built-in spline method for background estimation. The different background estimations by LCModel built-in spline function method might cause variations of metabolite concentrations. In addition, the relxation time (T2) corrections are only differential, since the T2 relaxation corrections involve the difference between in vivo and in vitro, but they can still be significant with long TE [8,31]. The basis set spectra are collected with a fully relaxed condition (TR = 10,000 ms), and a similar condition in in vivo is not feasible due to the time constraint. Hence, it is generally assumed that metabolites (in vivo) are relaxed back with the repetition time (TR = 3000 ms) in a clinical setting.

Previously, Schaller et al. [32] generated metabolite nulled macromolecule signals and imported them to a basis set. They reported a more robust baseline and improved fitting performance than the LCModel built-in spline method. In addition, they suggest that the inclusion of an experimentally measured macromolecule spectrum at 3T may provide good agreement with the literature values, especially GABA. Therefore, if we include simulated or measured macromolecule signals to our basis sets, we expect to remove negative correlations between metabolites and macromolecules to separate, for example, Gln, Lac, and NAAG, and improve quantification performances.

## 5. Conclusions

In this study, we compared the performances of the two basis sets. We found that the LCModel analysis with the in-house experimental basis set outperformed most metabolites and macromolecules. These differences in the performance of the two basis sets are due to line shape discrepancies. Such differences can be severe, especially for J-coupled metabolites. We believe that consistency can be improved by considering the localization effects, when simulating a basis set. From the correlation matrices, we can expect to quantify individually for Asp and NAAG, especially using the in-house experimental basis set. If we import the metabolite nulled signals to a basis set, we might improve the feasibility of separating metabolites, such as GABA, Lac, and Gln, in the prefrontal cortex of human brain at 3T. This may be useful for the quantification of brain metabolites (e.g., GABA, Glu, etc.) as imaging biomarkers of brain neurotransmission, oxidative stress, or inflammation, which are key targets of interest for central nervous system drug development.

## Figures and Tables

**Figure 1 metabolites-13-00368-f001:**
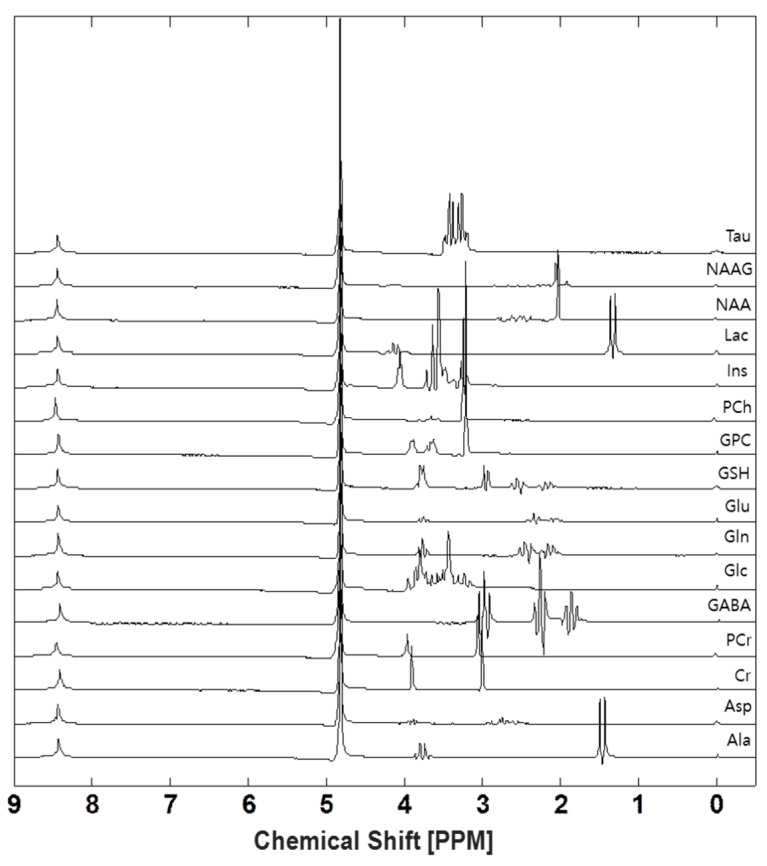
In-house experimentally acquired basis set generated using a phantom.

**Figure 2 metabolites-13-00368-f002:**
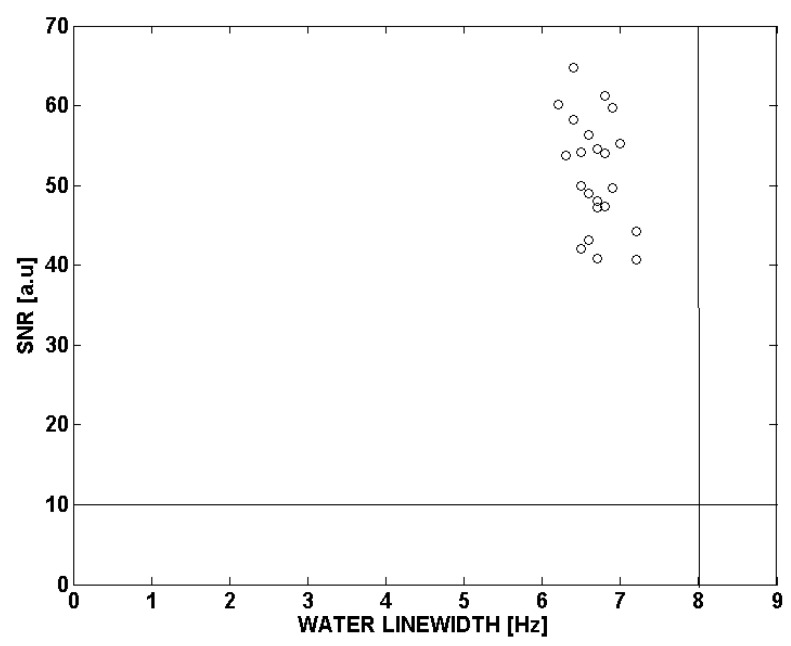
Overview of evaluated spectral qualities. Solid lines for FWHM (=8 Hz) and SNR (=10) criteria are drawn in the Figure. All 22 spectra are distributed in the satisfied region.

**Figure 3 metabolites-13-00368-f003:**
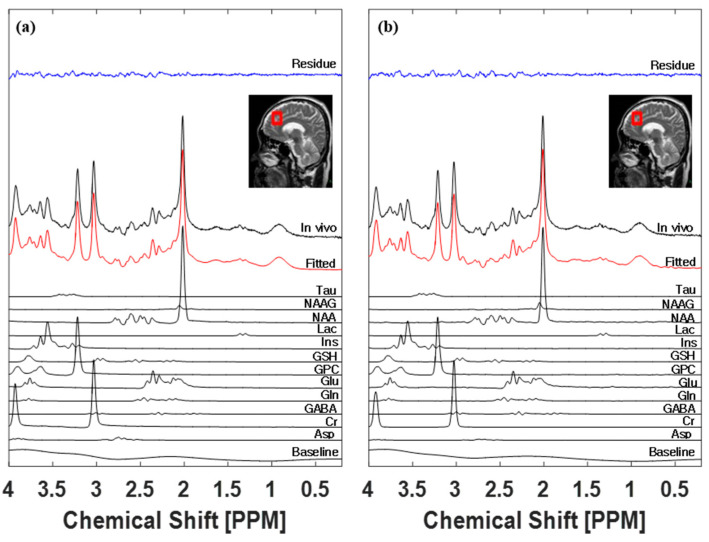
LCModel spectral fitting results with (**a**) and without (**b**) in-house basis sets. Fitting residue, VOI with sagittal T2 image, in vivo, fitted, individual spectra, and baseline are represented with names.

**Figure 4 metabolites-13-00368-f004:**
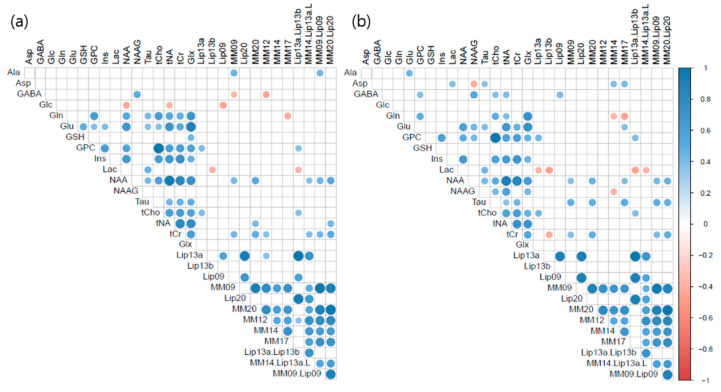
Correlation matrices of metabolite and macromolecule concentrations. Left, quantified by the in-house experimental basis set (**a**); Right, quantified by the provided simulated basis set (**b**).

**Figure 5 metabolites-13-00368-f005:**
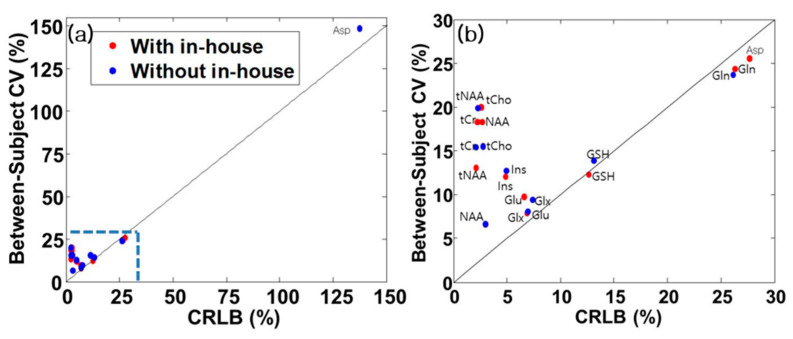
Coefficients of variation (CV) vs. CRLB and its zoomed Figure. (**a**) emphasizes that the CRLB of Asp using without in-house basis sets is remote from others. (**b**) shows that quantified metabolites are distributed within the CRLB < 35% and above.

**Table 1 metabolites-13-00368-t001:** CRLBs of metabolites and macromolecules and comparison of group difference by paired *t*-test.

CRLB [mM]						
Metabolite and Macromolecule	In-House		Simulated		In-House—Simulated	
	Mean	SE	Mean	SE	CI	*p*
Ala	0.08	0.02	0.10	0.02	[−0.06, 0.02]	0.25
Asp	0.33	0.004	0.42	0.02	[−0.13, −0.04]	*** <0.001
GABA	0.22	0.004	0.29	0.004	[−0.07, −0.06]	*** <0.001
Glc	0.05	0.02	0.07	0.02	[−0.05, 0.01]	0.24
Gln	0.37	0.008	0.43	0.007	[−0.07, −0.04]	*** <0.001
Glu	0.47	0.006	0.40	0.005	[−0.004, 0.02]	0.17
GSH	0.12	0.002	0.16	0.002	[−0.044, −0.038]	*** <0.001
Ins	0.14	0.002	0.19	0.003	[−0.05, −0.04]	*** <0.001
Lac	0.18	0.003	0.22	0.006	[−0.05, −0.04]	*** <0.001
NAA	0.13	0.006	0.20	0.004	[−0.05, −0.02]	*** <0.001
NAAG	0.12	0.006	0.18	0.005	[−0.08, −0.04]	*** <0.001
Tau	0.20	0.003	0.24	0.004	[−0.044, −0.035]	*** <0.001
tCho	0.02	0.0006	0.04	0.001	[−0.03, −0.02]	*** <0.001
tNA	0.14	0.004	0.16	0.005	[−0.04, −0.02]	*** <0.001
tCr	0.11	0.003	0.01	0.002	[0.004, 0.01]	** <0.01
Glx	0.53	0.01	0.56	0.008	[−0.05, −0.01]	*** <0.001
Lip13a	1.14	0.09	1.14	1.12	[−0.20, 0.18]	0.95
Lip13b	0.05	0.03	0.02	0.01	[−0.01, 0.08]	0.14
Lip09	0.32	0.06	0.41	0.05	[−0.20, 0.01]	0.09
MM09	0.74	0.02	0.74	0.02	[−0.03, 0.03]	0.99
Lip20	0.27	0.03	0.24	0.03	[0.0001, 0.06]	* <0.05
MM20	1.16	0.03	1.10	0.03	[0.04, 0.09]	*** <0.001
MM12	0.54	0.02	0.57	0.02	[−0.05, −0.01]	** <0.01
MM14	1.07	0.03	1.07	0.03	[−0.02, 0.03]	0.56
MM17	0.72	0.02	0.67	0.02	[0.02, 0.09]	** <0.01
Lip13a + Lip13b	1.11	0.09	1.14	0.11	[−0.18, 0.13]	0.70
MM14 + Lip13a + Lip13b + MM12	1.47	0.05	1.42	0.05	[−0.02, 0.11]	0.14
MM09 + Lip09	0.71	0.02	0.07	0.02	[−0.01, 0.03]	0.35
MM20 + Lip20	1.15	0.03	1.10	0.03	[0.03, 0.08]	*** <0.001

***** Mean CRLBs and standard error (SE) are presented. 95% confidence intervals of the mean difference between the groups are also presented. * *p* < 0.05, ** *p* < 0.01, *** *p* < 0.001.

**Table 2 metabolites-13-00368-t002:** Quantified concentrations and comparison of group difference by paired *t*-test.

Concentration [mM]						
Metabolite and Macromolecule	In-House		Simulated		In-House—Simulated	
	Mean	SE	Mean	SE	CI	*p*
Ala	0.06	0.02	0.05	0.02	[−0.03, 0.04]	0.58
Asp	1.24	0.07	0.68	0.09	[0.46, 0.67]	*** <0.001
GABA	0.47	0.03	0.89	0.04	[−0.48, −0.37]	*** <0.001
Glc	0.04	0.02	0.06	0.03	[−0.05, 0.003]	0.078
Gln	1.46	0.07	1.70	0.09	[−0.32, −0.17]	*** <0.001
Glu	6.26	0.11	5.84	0.09	[0.34, 0.51]	*** <0.001
GSH	0.94	0.03	1.20	0.03	[−0.29, −0.23]	*** <0.001
Ins	2.96	0.06	3.78	0.08	[−0.88, −0.78]	*** <0.001
Lac	0.33	0.03	0.42	0.03	[−0.12, −0.06]	*** <0.001
NAA	6.24	0.11	6.62	0.10	[−0.43, −0.32]	*** <0.001
NAAG	0.24	0.03	0.58	0.05	[−0.42, −0.26]	*** <0.001
Tau	0.52	0.04	0.81	0.05	[−0.33, −0.25]	*** <0.001
tCho	0.76	0.11	1.56	0.03	[−0.83, −0.77]	*** <0.001
tNA	6.49	0.03	7.21	0.13	[−0.76, −0.67]	*** <0.001
tCr	5.00	0.10	4.71	0.09	[0.26, 0.33]	*** <0.001
Glx	7.73	0.15	7.55	0.14	[0.05, 0.31]	** 0.0086
Lip13a	1.19	0.17	1.20	0.20	[−0.12, 0.08]	0.72
Lip13b	0.02	0.02	0.03	0.03	[−0.04, 0.02]	0.78
Lip09	0.10	0.02	0.19	0.03	[−0.13, −0.05]	*** <0.001
MM09	5.08	0.14	5.03	0.15	[0.004, 0.10]	* 0.034
Lip20	0.15	0.02	0.11	0.02	[0.02, 0.05]	*** <0.001
MM20	10.03	0.28	9.41	0.26	[0.48, 0.77]	*** <0.001
MM12	1.81	0.07	1.72	0.07	[0.07, 0.12]	*** <0.001
MM14	4.20	0.14	4.12	0.15	[−0.009, 0.16]	0.078
MM17	2.63	0.11	2.30	0.10	[0.28, 0.38]	*** <0.001
Lip13a + Lip13b	1.21	0.17	1.23	0.20	[−0.11, 0.07]	0.65
MM14 + Lip13a + Lip13b + MM12	7.22	0.27	7.07	0.28	[0.02, 0.28]	* 0.023
MM09 + Lip09	5.18	0.13	5.22	0.14	[−0.10, 0.01]	0.13
MM20 + Lip20	10.18	0.28	9.52	0.26	[0.51, 0.80]	*** <0.001

***** Mean absolute concentrations (mM) and standard error (SE) are presented. 95% confidence intervals of the mean difference between the groups are also presented. * *p* < 0.05, ** *p* < 0.01, *** *p* < 0.001.

**Table 3 metabolites-13-00368-t003:** Standard MRS Phantom LCModel quantification result.

Metabolite	Phantom Concentration	In-House		Simulated	
		Concentration	CRLB	Concentration	CRLB
Ins	7.5	6.10	0.30 (5)	7.89	0.37 (5)
Lac	5	3.37	0.30 (9)	4.33	0.39 (9)
Cho	3	2.45	0.07 (3)	3.54	0.11 (3)
NAA	12.5	11.58	0.23 (2)	13.33	0.27 (2)
Cr	10	9.85	0.30 (3)	10.37	0.21 (2)
Glu	12.5	11.85	0.71 (6)	11.76	0.71 (6)

Absolute concentration (mM) and CRLB are presented. Relative CRLB (=%SD) are also presented in the parenthesis.

## Data Availability

Data is unavailable due to privacy or ethical restrictions.

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
