# Peer review of "Experimental Basis Sets of Quantification of Brain 1H-Magnetic Resonance Spectroscopy at 3.0 T"

_metabolites, 2023, doi:10.3390/metabo13030368_

Round 1

Reviewer 1 Report

In the present manuscript, the Authors evaluate the performance of an in-house, experimentally measured basis set for localized in vivo MR spectra quantification and compare it with the performance of a vendor provided basis set at 3.0 T. In particular, the authors developed phantoms, one for each metabolite with reference chemicals, with adjusted temperature and pH in order to simulate physiological conditions. The experimental basis sets is then generated  by scanned each phantom by using an appropriate echo time and pulse sequence that can subsequently be applied to in vivo human data. A simulated basis set can be generated computationally using prior knowledge of physical and chemical characteristics of each metabolite and pulse sequence parameters. The simulation method is faster and easier ease than the experimental method. The authors found that fitting performances are better with the in-house basis set, which is more precise than the vendor provided basis set (a simulated basis set). In particular, aspartate (whose quantification was expected to be difficult also at 3.0T) is expected to have reliable CRLB (<30%) in the left prefrontal cortex of human brain at 3 T.

Overall, the paper is well written. The title describes the methodology but not summarizes the results. The introduction focusses on the problem, the limitations of the methods actually used and an overview on the successful improvements already published. The number of spectra included in the study is limited for this study and no mention about this limitation is in the discussion. Methodological details have been provided. Results are clearly described and the analyses and considerations seems to be appropriate.

The discussion shows the state of the art of the MR spectral analyses and compares them with the results obtained in the present work, but does not highlight the real added value of the present work. References are pertinent but they are not upgraded: references from the MRS study group of ISMRM should be included (for example the last one Marjańska et al, Magn Reson Med. 2022 should be added and a comment on it should be added in the discussion. The number of figures and tables is correct and they clearly summarize the results.

Strengths: Due to increasing application of MRS to diagnosis or prognosis of several illness, it is a good issue to reliable quantify brain metabolites and improving the presently used methods.  

Limitations:

The more important limitation relies on the fact that the authors never mention the T2 relaxation time in their considerations. They used a “short echo time” which is 35 ms but this echo time is not sufficiently “short” to ignore its effect on water and metabolite signals. A paragraph regarding these effects on the signals should be added in the discussion. The inclusion of ATTH20 and ATTmet in the method should be explained with more details giving the T2 of water and the metabolites at 3.0 T in the brain, in order to justify the given value in the calculation.

Moreover, phantom T2 should be made similar to human brain. No mention has been made about water or metabolite T2 of the phantom in the present manuscript.

Main issues

Considerations about the T2 loss of signals (water and metabolites) should be included in the present paper.

References should be upgraded and the works of the MRS study group of ISMRM should be mentioned.

Minor issues

Lines 187: “ATTH2O and ATTmet are set as 1 and 0.7 as default”- Be careful of the value that correspond to ATTH20 and ATTMet. It looks like they are reversed. Please rephrase to clarify.

Lines 371-372: The authors say: “Wilson et al. [16] suggested that if the concentration difference is in the ±0.99 mM range, then there are no significant differences between the basis sets”. It as no meaning to define a threshold in absolute values: if a metabolite has a concentration about 1-2 mM, differences about 0.99 mM are very large and smaller ones should be taken into consideration to define a better protocol for spectral analysis. A comment about this sentence should be added.

Lines 416-418:  The author wrote: “The output parameters ?? (8.4±19.08 vs. 62.5±77.0, mean±SD), ??(0.065±0.02 vs. 0.073±0.03), and ??/??(129±283.15 vs. 977±1572.06) were significantly lower with in-house experimental basis set (P<0.05)”. However, if we do the calculation, we found 62.5/0.073 = 856 and not 977. Please comment or correct it. 

Lines 471-475:  The authors wrote: “We could observe different averaged baselines between two approaches. 17.31% high baseline using with in-house basis sets (0.2-1.115 ppm), 55.53% high baseline using without in-house basis sets (1.115-3.01 ppm), 11.84% high baseline using in-house basis sets (3.01-3.515 ppm), and 2.51% high baseline using without in-house basis sets (3.515-4.0 ppm) were identified”. This sentence is not clear; please rephrase in order to clarify it. 

Line 477-478: “The authors say: “Therefore, our results have relatively large baseline variations”. It is not clear what the authors want to say. Which results? Is the large baseline variations amongst the spectra or between the two basis sets? Please clarify.

Author Response

Attached is the file.

Reviewer 2 Report

This article is about how to set up a better experimental basis to quantify human brain metabolites by H-Magnetic Resonance spectroscopy 3.0 T. Author introduced the methods of Basis sets and quantification. A total of 16 metabolite samples have been used to generate an experimental basis set in vitro, and H-MRS experiments were used to generate the measured basis sets in vivo. Some equations have been used for the quantification of metabolite concentrations. Figure 3. A and b, the peaks of GABA, Asp, Lac, and Gln were quite low and weak, and it was easy for them to be overlapped with other metabolites, so it should be difficult to check and quantify them, but the author told us this problem could be fixed by removing negative correlations between metabolites and macromolecules, including simulated or measured macromolecule signal to the in-house experimental basis set. It is an interesting article to explore the accurate method to measure metabolites and macromolecules in the human brain, but I still have a concern about it:

 1.       Author needs to add a little information about why to measure brain metabolites, and which kinds of brain diseases and misfunctions are accompanied by metabolic disorders. 

Author Response

Attached is the file.
